# Predictors of Cardiometabolic Health a Few Months Postpartum in Women Who Had Developed Gestational Diabetes

**DOI:** 10.3390/nu17030390

**Published:** 2025-01-22

**Authors:** Cristina Gómez Fernández, Rajna Golubic, Rea Mitsigiorgi, Tanvi Mansukhani, Josip Car, Kypros H. Nicolaides

**Affiliations:** 1Harris Birthright Research Centre for Fetal Medicine, King’s College, London SE5 8BB, UK; c.gomezfernandez@nhs.net (C.G.F.); rea.mitsigiorgi@nhs.net (R.M.); tanvi.mansukhani@nhs.net (T.M.); 2Faculty of Medicine, Complutense University of Madrid, 28040 Madrid, Spain; 3Oxford Centre for Diabetes, Endocrinology and Metabolism, Oxford OX3 9DU, UK; rajna.golubic@dtu.ox.ac.uk; 4Department of Women and Children’s Health, School of Life Course and Population Sciences, King’s College London, London SE5 8BB, UK; josip.car@kcl.ac.uk

**Keywords:** dysglycaemia, gestational diabetes, metabolic syndrome, postpartum clinic

## Abstract

Background: To assess the incidence of dysglycaemia and metabolic syndrome and factors associated with them 5 months postpartum in women with gestational diabetes mellitus (GDM) in their last pregnancy. Methods: We conducted an observational prospective cohort study in 558 women with previous GDM who attended a 5-month postpartum follow-up clinic. Backward elimination was performed to select significant factors for the multivariable logistic regression model. Dysglycaemia (prediabetes and type 2 diabetes (T2D)) and metabolic syndrome were used as outcomes in separate models. Results: Dysglycaemia was diagnosed in 202 (36.2%) women, including 174 (31.2%) with prediabetes and 28 (5.0%) with T2D. Women with dysglycaemia, compared with those with normoglycaemia, were more likely to be of black ethnicity (33.2 vs. 20.5%) and have severe GDM (31.7 vs. 16%), a higher postpartum BMI (29.5 vs. 27.6 kg/m^2^), and metabolic syndrome (20 vs. 7%). Multivariable logistic regression analysis showed that significant predictors of dysglycaemia were black (OR 2.09; 95% CI: 1.27–3.46) and mixed ethnicity (OR 3.05; 95% CI: 1.26–7.42), diagnosis of GDM before 24 weeks gestation (OR 3.05, 95% CI: 1.90–4.91), and treatment of GDM with metformin (OR 1.63; 95% CI: 1.05–2.55) or insulin (OR 2.08; 95% CI: 1.14–3.79) rather than diet alone. Significant predictors of metabolic syndrome were postpartum maternal BMI (OR 5.49; 95% CI: 2.60–11.59) and absence of breastfeeding (OR 2.14; 95% CI: 1.21–3.77). Conclusions: At 5 months postpartum, a high proportion of women who developed GDM showed evidence of dysglycaemia. Future studies should investigate interventions that could reduce the risk of short- and long-term consequences of suboptimal cardiometabolic health in such women.

## 1. Introduction

Gestational diabetes mellitus (GDM), defined as impaired glucose tolerance first diagnosed during pregnancy [1], has a global prevalence of 14% [2]. Shortly after delivery, glucose homeostasis is restored to non-pregnancy levels, but women who suffer from GDM during pregnancy have a ten-fold higher risk of future dysglycaemia (defined as having prediabetes or type 2 diabetes (T2D)) compared with women with normoglycaemia in pregnancy [3]. Dysglycaemia has been reported to be as high as 26% in the postpartum period [4]. This increased risk is most pronounced in the first five years postpartum [5]. Additionally, women who developed GDM have a 2.3-fold increase in cardiovascular events in the first decade postpartum [6]. Given the highest risk of T2D is 5 years postpartum, women with previous GDM tend to develop T2D relatively early in their lives. This, in turn, means a longer duration of T2D and a higher lifetime risk of cardiovascular disease and associated complications [7].

According to current clinical practice guidance, women who have GDM are advised to assess fasting plasma glucose or glycated haemoglobin A1c (HbA1c) at 4 to 12 weeks postpartum [8,9,10,11]. However, the reported rates of attendance for such follow-up are low, ranging from 23% to 58% [12]. Furthermore, there is no consensus on long-term follow-up or whether the assessment of other cardiometabolic risk factors should be introduced. In our centre, we have established a routine 5-month follow-up clinic for women who developed GDM, with the aim of monitoring their cardiometabolic health and postnatal outcomes longitudinally.

The objectives of this study were to quantify the incidence of dysglycaemia and metabolic syndrome in the postnatal clinic and determine their predictors.

## 2. Materials and Methods

### 2.1. Study Design and Participants

At the Foetal Medicine Unit of the King’s College Hospital, London, women attend routine foetal ultrasound examinations at 12, 20, and 36 weeks of gestation. Data on pregnancy outcomes are collected from hospital maternity records or from general practitioners providing primary care for the women. Every woman who developed GDM was asked to attend a postnatal clinic.

During a 12-month period (September 2023 to August 2024), we invited all 704 women who developed GDM to attend this postnatal clinic and 558 (79.3%) did so at a median of 5.0 (IQR 4.4–7.0) months after delivery. Women gave written informed consent to participate in the study, which was approved by the National Health Service (NHS) Research Ethics Committee (reference 18/NI/0013; Integrated Research Application System ID 237936).

At the postnatal visit, we recorded maternal demographic characteristics, obstetric and medical history (age, self-reported ethnicity, family history of DM, medications related to increased risk of hyperglycaemia and/or antihypertensive medication, gravidity, parity, method of conception, GDM in previous pregnancy, body mass index (BMI) at 11–13 and 36 weeks gestation), and details of pregnancy outcome (gestational age at diagnosis of GDM, treatment required for GDM control, gestational age at delivery, birthweight and birthweight centile at delivery). Additionally, we measured the weight (kg) and height (m) of the women and calculated their BMI, systolic and diastolic blood pressure (mmHg), and upper arm and waist circumference (cm). We performed a 75 g oral glucose tolerance test (OGTT) and obtained venous blood for measurement of HbA1c (mmol/mol), serum triglycerides (mmol/L), and serum high-density lipoprotein cholesterol (mmol/L). OGTT was performed in accordance with the WHO guidelines using a glucose load containing the equivalent of 75 g of anhydrous glucose dissolved in water. All laboratory assays were performed according to standard operating procedures.

### 2.2. Outcome Measures

The outcome measures were dysglycaemia and metabolic syndrome. Dysglycaemia was defined as having prediabetes or diabetes. Prediabetes was defined according to the criteria of the National Institute for Health and Care Excellence (NICE) [13] as HbA1c between 42 and 47 mmol/mol or fasting plasma glucose between 5.5 and 6.9 mmol/L or impaired glucose tolerance (2 h blood glucose during OGTT) between 7.8 and 11 mmol/L. T2D was defined using the World Health Organization criteria as HbA1c ≥48 mmol/mol or fasting plasma glucose ≥7.0 mmol/L or 2 h blood glucose during OGTT of ≥11.1 mmol/L or random plasma glucose is ≥11.1 mmol/L in a patient with symptoms of hyperglycaemia [14]. Fasting was defined as no caloric intake for at least 8 h.

Metabolic syndrome was diagnosed if at least 3 of the 5 following criteria were fulfilled [15]: central obesity defined as waist circumference ≥88 cm, serum triglycerides ≥1.7 mmol/L or use of triglyceride-lowering treatment, serum high-density lipoprotein cholesterol <1.3 mmol/L or use of cholesterol-lowering treatment, blood pressure ≥130/85 mmHg or antihypertensive treatment, and fasting plasma glucose ≥5.5 mmol/L. Ethnicity-specific criteria were applied.

### 2.3. Statistical Analysis

Quantitative variables were presented as mean and standard deviation (SD) or median and interquartile range (IQR) if there was a departure from a normal distribution. Continuous variables were compared between normoglycemic women and those with dysglycaemia using the Student’s t-test or median test. Differences in the frequencies of categorical variables between those with normoglycaemia and dysglycaemia were tested using the chi-square test. Quantitative variables were categorised by the median value of the whole cohort.

Backward multivariable logistic regression was performed to assess which factors were predictive of dysglycaemia and metabolic syndrome (outcomes in separate models). Prior to the regression analysis, continuous variables, such as age and BMI, were centred by subtracting the median from each value. In the first step, univariate analysis was performed, including maternal demographic characteristics, variables from the medical history and pregnancy outcome, breastfeeding, and use of medications associated with an increased risk of diabetes. In the second step, multivariate logistic regression was carried out using the variables with *p* < 0.1 in the univariate analysis. The relative effect of each variable was calculated as odds ratio (OR) and its 95% confidence interval (95% CI). The calibration of the model was evaluated by the Hosmer–Lemeshow test and its discrimination by receiver operating characteristic (ROC) curves. We conducted sensitivity analyses using the definition of prediabetes according to the American Diabetes Association (ADA) [1].

All statistical tests were two-sided, and *p* < 0.05 was considered statistically significant. The statistical software SPSS ver.29 was used for the analyses.

## 3. Results

The characteristics of the study population are summarised in Table 1 and Appendix A. After a median postpartum follow-up of 5 months, dysglycaemia was diagnosed in 202 (36.2%) of the 558 participants, including 174 (31.2%) with prediabetes and 28 (5.0%) with T2D. In women with dysglycaemia, compared to the normoglycaemic group, there was a higher prevalence of the following characteristics: black ethnicity (33.2% vs. 20.5%), GDM in a previous pregnancy (40.2% vs. 23.9%), use of insulin for the treatment of GDM (31.7% vs. 16%), diagnosis of GDM at <24 weeks gestation (36.5% vs. 11.7%), and higher BMI at the postnatal visit (29.5 kg/m^2^; IQR 24.8–34.3 vs. 27.6 kg/m^2^; IQR 23.6–31.6). The median birthweight percentile of the neonates of women with dysglycaemia tended to be lower compared to the neonates of normoglycaemic women (39.6 (IQR 20.6–71.0) vs. 56.1 (IQR 22.5–76.4)).

The median (IQR) values for the components of metabolic syndrome are shown in Table 2. The criteria for metabolic syndrome were fulfilled by 67 (12.0%) of 558 women, including 26 (7.3%) in the normoglycaemia and 41 (20.3%) in the dysglycaemia groups (Table 2). In the dysglycaemia group, there was a significantly higher prevalence of central obesity (67.3% vs. 53.9%), plasma triglycerides ≥1.7 mmol/L or triglyceride-lowering treatment (18.8% vs. 8.7%), and fasting plasma glucose ≥5.5 mmol/L (21.8% vs. 0%). Additionally, compared to the normoglycaemic group, women in the dysglycaemia group had significantly higher systolic blood pressure (117.5 mmHg vs. 115.1 mmHg), HbA1c concentration (42 mmol/mol; IQR 38–44 vs. 37 mmol/mol; IQR 35–39), and 2 h plasma glucose level after a 75 g OGTT (7.1 mmol/L; IQR 5.8–8.5 vs. 5.5 mmol/L; IQR 4.7–6.2).

The predictors of dysglycaemia and metabolic syndrome from the logistic regression analysis are shown in Table 3 and Table 4, respectively. Regarding dysglycaemia (Table 3), of the considered variables from the univariate analysis (ethnicity, GDM in previous pregnancy, gestational age at delivery, birthweight percentile, treatment required for GDM, gestational age at diagnosis of GDM, BMI at the postnatal visit, waist circumference, central obesity ≥ 88 cm, serum triglycerides ≥ 1.7 mmol/L or triglyceride-lowering treatment, serum HDL-cholesterol, and blood pressure ≥130/85 mmHg or antihypertensive treatment), predictors of dysglycaemia were black (OR 2.09; 95% CI: 1.27–3.46) and more than one ethnicity (OR 3.05; 95% CI: 1.26–7.42), diagnosis of GDM at <24 weeks gestation (OR 3.05, 95% CI: 1.90–4.91), need for metformin (OR 1.63; 95% CI: 1.05–2.55) or insulin (OR 2.08; 95% CI: 1.14–3.79) for glycaemic control of GDM, birthweight percentile (OR 1.96; 95% CI: 1.30–2.95), and serum triglycerides (mmol/L) ≥ 1.7 mmol/L or triglyceride-lowering treatment (OR 2.29; 95% CI: 1.29–4.07). Early gestational age at diagnosis (<24 weeks as defined at the first summit discussing gestational diabetes diagnosed early in pregnancy [16]) was a risk factor for dysglycaemia (OR 3.05; 95% CI: 1.90–4.91).

**Table 2 nutrients-17-00390-t002:** Components for the diagnosis of metabolic syndrome in women with normoglycaemia and dysglycaemia.

	All *n* = 558	Normoglycaemia *n* = 356	Dysglycaemia *n* = 202	*p*-Value
**Metabolic syndrome**	67 (12.0)	26 (7.3)	41 (20.3)	<0.001
Waist circumference (cm)	91.3 (82–101)	89 (82–98.6)	95.8 (83.7–104.8)	<0.001
**Central obesity ≥ 88 cm**	328 (58.8)	192 (53.9)	136 (67.3)	0.004
Serum triglycerides (mmol/L)	0.9 (0.6–1.3)	0.8 (0.6–1.2)	0.9 (0.7–1.4)	0.002
**≥1.7 mmol/L or** **triglyceride-lowering treatment**	69 (12.4)	31 (8.7)	38 (18.8)	0.001
Serum HDL-Cholesterol (mmol/L)	1.5 (1.3–1.8)	1.5 (1.3–1.9)	1.5 (1.2–1.7)	0.067
**<1.3 mmol/L or cholesterol-lowering treatment**	127 (22.8)	74 (20.8)	53 (26.2)	0.147
Systolic blood pressure (mmHg)	116.5 (109.8–123.6)	115.1 (108.8–123.5)	117.5 (111.0–124.3)	0.021
Diastolic blood pressure (mmHg)	74.3 (69.3–79.8)	73.6 (68.1–79.2)	74.8 (70.4–80.8)	0.087
**Blood pressure ≥130/85 mmHg or antihypertensive treatment**	97 (17.4)	54 (15.2)	43 (21.3)	0.076
Fasting plasma glucose (mmol/L)	4.5 (4.2–4.9)	4.5 (4.2–4.8)	4.8 (4.4–5.3)	<0.001
**≥5.5 mmol/L**	44 (7.9)	0 (0)	44 (21.8)	<0.001
75 g OGTT: plasma glucose at 120 min (mmol/L)	5.9 (4.9–7.0)	5.5 (4.7–6.2)	7.1 (5.8–8.5)	<0.001
Haemoglobin A1c (mmol/mol)	38 (36–41)	37 (35–39)	42 (38–44)	<0.001

Quantitative variables expressed as median and IQR (p25-p75). Qualitative variables expressed as number of cases (%). Ethnicity-specific cut-offs were used [17].

Regarding metabolic syndrome (Table 4), BMI at 11–13 and 36 weeks of gestation and in the postnatal visit, family history of diabetes, GDM in a previous pregnancy, birthweight percentile, treatment of GDM, gestational age at diagnosis of GDM, breastfeeding and 75 g OGTT plasma glucose at 120 min were considered variables in the univariate analysis. When performing multivariate analysis, predictors of metabolic syndrome were the need for metformin for glycaemic control of GDM (OR 2.17; 95% CI: 1.06–4.42), maternal BMI at the postpartum visit (OR 5.49; 95% CI: 2.60–11.59), and absence of breastfeeding (OR 2.14; 95% CI: 1.21–3.77).

As shown in Appendix A, we performed a sensitivity analysis using the ADA definition of prediabetes (HbA1c 39–48 mmol/mol). We found to be predictors for dysglycaemia black (OR 3.71; 95% CI: 2.26–6.10), South Asian (OR 1.67; 95% CI: 1.00–2.78), and East Asian (OR 3.02; 95% CI: 1.47–6.19) ethnicities; birthweight percentile (OR 1.57; 95% CI: 1.06–2.32), and diagnosis of GDM at <24 weeks gestation (OR 2.09; 95% CI: 1.23–3.56).

## 4. Discussion

### 4.1. Main Findings of the Study

In this large study of women from an inner-city population who had developed GDM, follow-up at a median of 5 months postpartum showed that 36% of women had dysglycaemia, including 31% with prediabetes and 5% with T2D. In women with dysglycaemia, compared to those with normoglycaemia, there was a higher prevalence of black ethnicity, a severe form of GDM (i.e., diagnosis at <24 weeks and treatment with insulin), as well as higher body mass index at the postnatal visit. Additionally, in the dysglycaemia group, the prevalence of metabolic syndrome was substantially higher than in the normoglycaemic group (20% vs. 7%, *p* < 0.001). In a logistic regression analysis, predictors of dysglycaemia were black and more than one ethnicity, diagnosis of GDM at <24 weeks gestation, and need for metformin or insulin for glycaemic control of GDM. Predictors of metabolic syndrome were the need for insulin therapy for glycaemic control of GDM, maternal BMI at the postpartum visit, and absence of breastfeeding.

### 4.2. Comparison with Previous Studies

A meta-analysis of 15 studies of postpartum assessment in 4560 women with previous GDM reported that the incidence of dysglycaemia ranged from 9 to 63% [18]. Such a wide range of results reflects the large differences between studies, which included from 103 to 1010 patients examined at 1 month to 20 years after delivery and used different criteria for diagnosis of the condition. Another large meta-analysis of 20 studies [3], including a total of 1,332,373 individuals (67,956 women with GDM and 1,264,417 controls), reported that the relative risk of T2D in women with previous GDM vs. controls with no GDM history was 17.1 (95% CI: 8.95–32.55) in studies with follow-ups of 1–5 years, 10.4 (95% CI: 5.68–19.11) in those with follow-ups of between 5 and 10 years, and 8.1 (95% CI: 4.34–15.08) in those with follow-ups of more than 10 years.

We found that women of black ethnicity had a 2-fold higher risk of developing dysglycaemia than white women. This is consistent with the results of a Danish nationwide registry of 20,873 women with GDM who were followed up for a mean of 7.3 years; 10.9% of the women developed T2D, and the risk was particularly high in-migrant women, especially of black and South Asian ethnicity, compared to Danish women [19].

Regarding maternal characteristics, we could not find any differences in maternal age between the groups. However, the median maternal age (35.6, IQR 32.0–38.2 years) appears to be higher compared to other studies. As a well-known risk factor for the development of short- and long-term complications after pregnancy, this may have an impact on the incidence of dysglycaemia in our cohort. Other studies [20] have reported that elevated maternal age is an independent risk factor for dysglycaemia and metabolic syndrome postpartum. Maternal age has also been associated with foetal macrosomia.

We also noted that the prevalence of dysglycaemia was higher in women who required insulin therapy for GDM and in those where the diagnosis of GDM was made at <24 weeks gestation. This is consistent with the results of previous studies, which reported that pregnancies with early diagnosis of GDM are at greater risk of pregnancy complications and the need for insulin treatment [21]. Additionally, the early GDM phenotype has a higher severity and morbidity risk affecting the development of multiple foetal tissues, with short-term and long-term consequences [22].

In our study, 20% of the patients with dysglycaemia met the criteria for metabolic syndrome, which was substantially higher when compared to the normoglycaemic group (7%). These results were also found by a Swiss study (1261 women divided into 1185 women with classical GDM diagnosed between 24 and 28 weeks and 76 women with early GDM diagnosed before 20 weeks), which showed that at 6–8 weeks postpartum, those with early compared to late diagnosis of GDM had a more atherogenic lipid profile, a higher prevalence of metabolic syndrome, and higher prevalence of pre-diabetes and T2D [23]. Additionally, in our cohort, women with metabolic syndrome had higher systolic blood pressure than those with dysglycaemia (117.5 mmHg vs. 115.1 mmHg). However, although significant, this result has no clinical relevance.

Some studies have reported logistic regression models for the prediction of dysglycaemia in postpartum GDM. However, the studies varied in diagnostic criteria, the interval from delivery to follow-up, and in the predictive factors included in the analysis. A recently published retrospective cohort study in Singapore of 942 patients, examined at 6 to 12 weeks after delivery, reported that the incidence of dysglycaemia was 16.7%, including 13.2% with prediabetes and 3.5% with T2DM [24]. In the dysglycaemia, compared to the normoglycaemia group, there was a higher mean age and proportion of primiparous women but no significant difference in ethnicity, body mass index at first visit (6 weeks postpartum), gestational age at delivery, or birth weight. Although these results differ from ours, this is likely to be explained by the ethnicity of women included in the study (45.4% Chinese, 21.7% Malay, and 14.3% Indian). A meta-analysis of 39 studies, including 95,750 women diagnosed with GDM and duration of follow-up of between 6 weeks and 20 years postpartum, reported similar results to those of our cohort in terms of increased risk of dysglycaemia in women of non-white ethnicity, early diagnosis of GDM, and use of insulin therapy [25]. However, the study reported additional factors to be predictive of dysglycaemia, including high BMI, family history of diabetes, advanced maternal age, multiparity, hypertensive disorders in pregnancy, preterm birth, and diagnosis of GDM at <24 weeks gestation.

We have previously reported that at 35–36 weeks gestation, patients with GDM, compared to controls with uncomplicated pregnancies, have reduced left ventricular diastolic and systolic functional indices; additionally, when the cardiac function of these women was assessed at 6 months postpartum, there was a lower degree of improvement in left ventricular myocardial relaxation and in global longitudinal systolic strain despite optimal glucose management during pregnancy [26]. In another study, we found that in women with GDM examined at 26–40 weeks of gestation, there was evidence of subclinical reduction in biventricular systolic function [27].

Although we did not include echocardiographic assessment in this study, identifying patients at risk of postpartum dysglycaemia and metabolic syndrome after GDM would allow for evaluation of cardiac function to prevent short- and long-term consequences. Additionally, further larger interventional studies should be carried out to understand the implications of GDM, postpartum dysglycaemia, and cardiac remodelling.

Strategies to reduce postpartum cardio-metabolic risk in women with previous GDM should be personalised and integrate pharmacotherapy and lifestyle modification depending on a woman’s risk profile and co-morbidities. However, before such a multi-level and multi-disciplinary approach is implemented in practice, further well-designed research is required to establish the efficacy of the combined use of pharmacotherapy and lifestyle interventions and its effectiveness and cost-effectiveness in the “real world” while addressing implementation challenges at the level of individuals and systems [28].

Pharmacotherapies investigated to prevent T2D or improve metabolic health postpartum in women with previous GDM include metformin, pioglitazone, dapagliflozin, and liraglutide [28]. Metformin and pioglitazone [29,30] were associated with an up to 55% decreased risk of T2D compared to standard of care over a period of up to 3 years. Dapagliflozin with metformin or liraglutide with metformin [31,32] were significantly more effective in improving fasting plasma glucose, lipid profile, and weight than metformin monotherapy. Other incretin-based treatments have not yet been investigated in women with previous GDM, and their long-term efficacy and safety have not been established in this group.

### 4.3. Strengths and Weaknesses

The strengths of our study include a large sample size of an ethnically heterogeneous population with a high proportion of women of black and South Asian ethnicity, which are the groups with a greater risk of developing T2D and cardiometabolic disorders, thus increasing the likelihood of our results being generalizable to these groups. Additionally, a high response rate decreased the probability of selection bias.

Several limitations need to be considered when interpreting our findings. Firstly, it is possible that the calculated odds ratios of dysglycaemia and metabolic syndrome might have been underestimated, given that interventions in routine clinical practice might have taken place (i.e., diet modification during pregnancy). Secondly, the possibility of residual confounding due to unmeasured variables cannot be excluded. For example, we did not assess physical activity and diet, which are the factors that play an important role in reducing the risk of dysglycaemia and metabolic syndrome. Other potential unmeasured factors include sleep and psychological stress. Thirdly, this was an observational study, and causal associations cannot be inferred.

Given the ethnic composition of our sample, the findings may not be fully generalizable to predominantly White women. However, our results provide evidence of the factors associated with increased risk of dysglycaemia and metabolic syndrome in ethnic minorities (predominantly Black women), thus calling for culturally tailored interventions to address the modifiable risk factors. Such examples would include dietary interventions tailored to food items consumed among ethnic groups and patterns of physical activity that are deemed acceptable and feasible to adhere to in these subgroups of women.

## 5. Conclusions

A high proportion of women who had developed GDM showed evidence of dysglycaemia at a median postpartum follow-up of 5 months. Future studies should investigate interventions that could potentially reduce the risk of short- and long-term consequences of suboptimal cardiometabolic health in such women.

## Figures and Tables

**Table 1 nutrients-17-00390-t001:** Baseline characteristics of the study population.

	All *n* = 558	Normoglycaemia *n* = 356 (63.8%)	Dysglycaemia *n* = 202 (36.2%)	*p*-Value
**Demographic characteristics**				
Age (years) *	35.6 (32–38.2)	35.4 (32–38.1)	35.8 (32.0–38.3)	0.290
Ethnicity				<0.001
White	244 (43.7)	180 (50.6)	64 (31.7)	
Black	140 (25.1)	73 (20.5)	67 (33.2)	
South Asian	99 (17.7)	61 (17.1)	38 (18.8)	
East Asian	48 (8.6)	29 (8.1)	19 (9.4)	
More than one	27 (4.8)	13 (3.7)	14 (6.9)	
First or second-degree family history of diabetes	276 (49.5)	169 (47.5)	107 (53)	0.384
Parity				0.083
Nulliparous	234 (41.9)	159 (44.7)	75 (37.1)	
Parous	324 (58.1)	197 (55.3)	127 (62.9)	
Parous with previous GDM	92 (28.4)	47 (23.9)	51 (40.2)	0.001
Method of conception				0.404
Spontaneous	495 (88.7)	311 (87.4)	184 (91.1)	
Ovulation induction	4 (0.7)	3 (0.8)	1 (0.5)	
In vitro fertilisation	59 (10.6)	42 (11.8)	17 (8.4)	
BMI at 11–13 weeks gestation (kg/m^2^)	27.9 (23.8–33.0)	27.6 (23.5–32.5)	28.9 (24.3–34.3)	0.053
BMI at 36 weeks (kg/m^2^)	31.0 (27.2–35.2)	30.9 (27.0–34.6)	31.6 (27.5–36.4)	0.361
**Pregnancy outcome**				
Gestational age at delivery (weeks)	39.0 (38.3–39.6)	39.1 (38.4–39.7)	38.9 (38.0–39.4)	0.025
Birthweight percentile	49.6 (21.1–74.8)	56.1 (22.5–76.4)	39.6 (20.6–71.0)	0.001
Treatment of GDM				<0.001
Diet	209 (37.5)	161 (45.2)	48 (23.8)	
Metformin	228 (40.9)	138 (38.8)	90 (44.6)	
Insulin (+/− metformin)	121 (21.7)	57 (16)	64 (31.7)	
Gestational age at diagnosis of GDM				<0.001
<24 weeks	113 (20.7)	41 (11.7)	72 (36.5)	
≥24 weeks	433 (79.3)	308 (88.3)	125 (63.5)	
**Postnatal visit**				
Body mass index (kg/m^2^)	28.2 (24.0–32.8)	27.6 (23.6–31.6)	29.5 (24.8–34.3)	0.007
Antihypertensive medication	7 (1.3)	2 (0.6)	5 (2.5)	0.059
Medications associated with an increased risk of hyperglycaemia *	19 (3.4)	12 (3.4)	7 (3.5)	0.953

Quantitative variables expressed as median and IQR (p25-p75). Qualitative variables expressed as number of cases (%). * Medications associated with increased risk of hyperglycaemia were systemic glucocorticoids, selective serotonin reuptake inhibitors, antipsychotics, and antiretrovirals.

**Table 3 nutrients-17-00390-t003:** Predictors of dysglycaemia.

Predictor	Univariable		Multivariable	
	OR (95% CI)	*p*-Value	OR (95% CI)	*p*-Value
**Demographic characteristics**				
Age ≥36 years	1.12 (0.79–1.58)	0.525		
Ethnicity				
White (reference)	1.00			
Black	2.58 (1.67–4.00)	<0.001	2.09 (1.27–3.46)	0.004
South Asian	1.75 (1.07–2.88)	0.027	1.43 (0.82–2.49)	0.204
East Asian	1.84 (0.97–3.51)	0.063	1.86 (0.87–3.91)	0.101
Mixed	3.03 (1.35–6.79)	0.007	3.05 (1.26–7.42)	0.014
First or second-degree family history of diabetes (reference: no family history)	1.25 (0.88–1.76)	0.212		
Parity				
Parous (reference: nulliparous)	1.07 (0.73–1.59)	0.719		
Previous GDM (reference: parous without previous GDM)	2.30 (1.42–3.73)	<0.001		
Method of conception				
Spontaneous (reference)	1.00			
Ovulation induction	0.56 (0.06–5.46)	0.620		
In vitro fertilisation	0.68 (0.38–1.24)	0.209		
BMI at 11–13 weeks gestation ≥28 kg/m^2^	1.41 (1.00–2.00)	0.051		
BMI at 36 weeks ≥31 kg/m^2^	1.19 (0.84–1.68)	0.329		
**Pregnancy outcome**				
Gestational age at delivery ≥39 weeks	1.56 (1.10–2.21)	0.012		
Birthweight percentile ≥50 percentile	1.92 (1.35–2.74)	<0.001	1.96 (1.30–2.95)	0.001
Treatment of GDM				
Diet (reference)	1.00			0.030
Metformin	2.40 (1.60–3.59)	<0.001	1.63 (1.05–2.55)	0.031
Insulin (+/− metformin)	3.60 (2.11–6.14)	<0.001	2.08 (1.14–3.79)	0.018
Gestational age at diagnosis of GDM <24 weeks	4.33 (2.80–6.69)	<0.001	3.05 (1.90–4.91)	<0.001
**Postnatal visit**				
Postnatal BMI ≥28 kg/m^2^	1.68 (1.19–2.39)	0.004		
Breastfeeding (reference: yes)	1.10 (0.77–1.56)	0.612		
Antihypertensive medication (reference: no)	4.49 (0.86–23.37)	0.074		
Medications associated with an increased risk of diabetes (reference: no)	1.03 (0.40–2.66)	0.953		
Waist circumference ≥91 cm	1.85 (1.30–2.64)	<0.001	1.52 (0.98–2.34)	0.062
Serum triglycerides ≥0.9 mmol/L	1.50 (1.06–2.12)	0.023		
Serum HDL-cholesterol <1.5 mmol/L	1.48 (1.04–2.09)	0.029		
Systolic blood pressure ≥117 mmHg	1.52 (1.08–2.16)	0.017		
Diastolic blood pressure ≥74 mmHg	1.32 (0.93–1.86)	0.120		

Continuous variables were dichotomized as lower than the median and greater than or equal to the median. The multivariable model contains only those variables which had *p* < 0.1 in the backward elimination.

**Table 4 nutrients-17-00390-t004:** Predictors of metabolic syndrome.

Predictor	Univariable		Multivariable	
	OR (95% CI)	*p*-Value	OR (95% CI)	*p*-Value
**Demographic characteristics**				
Age ≥36 years	0.88 (0.53–1.48)	0.631		
Ethnicity				
White (reference)	1.00	0.269		
Black	0.59 (0.30–1.16)	0.128		
South Asian	0.80 (0.40–1.62)	0.537		
East Asian	0.26 (0.06–1.10)	0.067		
Mixed	1.00 (0.33–3.06)	>0.999		
First or second-degree family history of diabetes (reference: no family history)	1.82 (1.07–3.07)	0.026	1.63 (0.91–2.90)	0.098
Parity				
Parous (reference: nulliparous)	1.31 (0.72–2.37)	0.377		
Previous GDM (reference: parous without previous GDM)	2.15 (1.10–4.21)	0.026		
Method of conception				
Spontaneous (reference)	1.00			
Ovulation induction	Undefined			
In vitro fertilisation	0.97 (0.42–2.23)	0.937		
BMI at 11–13 weeks gestation ≥28 kg/m^2^	4.14 (2.27–7.55)	<0.001		
BMI at 36 weeks ≥31 kg/m^2^	3.41 (1.91–6.08)	<0.001		
**Pregnancy outcome**				
Gestational age at delivery ≥39 weeks	1.55 (0.92–2.59)	0.099		
Birthweight percentile >50 percentile	0.56 (0.33–0.96)	0.033	0.68 (0.38–1.20)	0.100
Treatment of GDM				
Diet (reference)	1.00		1.00	
Metformin	2.74 (1.43–5.27)	0.002	2.17 (1.06–4.42)	0.034
Insulin (+/− metformin)	2.77 (1.23–6.27)	0.014	2.10 (0.87–5.06)	0.100
Gestational age at diagnosis of GDM < 24 weeks	2.02 (1.15–3.57)	0.015		
**Postnatal visit**				
Postnatal BMI ≥28 kg/m^2^	7.35 (3.56–15.17)	<0.001	5.49 (2.60–11.59)	<0.001
Breastfeeding (reference: yes)	2.32 (1.38–3.89)	0.001	2.14 (1.21–3.77)	0.009
Antihypertensive medication (reference: no)	1.22 (0.14–10.25)	0.858		
Medications associated with an increased risk of diabetes (reference: no)	0.42 (0.06–3.20)	0.402		
75 g OGTT: plasma glucose at 120 min ≥5.9 mmol/L	2.03 (1.18–3.51)	0.011		
Haemoglobin A1c ≥38 mmol/mol	1.42 (0.84–2.40)	0.193		

Continuous variables were dichotomized as lower than the median and greater than or equal to the median. The multivariable model contains only those variables which had *p* < 0.1 in the backward elimination.

## Data Availability

The original contributions presented in this study are included in the article/Appendix A. Further inquiries can be directed to the corresponding author.

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
