# Peer review of "Predictors of Cardiometabolic Health a Few Months Postpartum in Women Who Had Developed Gestational Diabetes"

_nutrients, 2025, doi:10.3390/nu17030390_

Round 1

Reviewer 1 Report

Comments and Suggestions for Authors

In this study conducted on 558 participants, the authors found that the main predictors of dysglycaemia were black and more than one ethnicity, diagnosis of GDM <24 weeks gestation and need for metformin or insulin for glycaemic control of GDM. Predictors of metabolic syndrome were need for insulin therapy for glycaemic control of GDM, maternal weight at the postpartum visit and absence of breastfeeding. 

The authors could also discuss the pathophysiological consequences of persistent dysglycaemia and/or metabolic syndrome in pregnancies complicated by GDM.

I suggest the authors to expand the Discussion section by discussing the cardiac remodeling associated with GDM. Literature data indicate that GDM cardiac remodeling is characterized by an increase in left ventricular mass and stiffness, leading to an early decline in diastolic function and concurrent changes in myocardial deformation indices, which commonly precede systolic dysfunction. Importantly, these changes may be persistent in postpartum period, particularly in GDM women with obesity and uncontrolled diabetes. The authors could mention and discuss the following papers: PMID: 32530101, PMID: 33949740 and PMID: 37951113

Author Response

Responses to comments of the reviewers

We enclose a revised manuscript and responses to the constructive comments of the reviewers.

REVIEWER #1

“In this study conducted on 558 participants, the authors found that the main predictors of dysglycaemia were black and more than one ethnicity, diagnosis of GDM <24 weeks gestation and need for metformin or insulin for glycaemic control of GDM. Predictors of metabolic syndrome were need for insulin therapy for glycaemic control of GDM, maternal weight at the postpartum visit and absence of breastfeeding.

The authors could also discuss the pathophysiological consequences of persistent dysglycaemia and/or metabolic syndrome in pregnancies complicated by GDM.

I suggest the authors to expand the Discussion section by discussing the cardiac remodeling associated with GDM. Literature data indicate that GDM cardiac remodeling is characterized by an increase in left ventricular mass and stiffness, leading to an early decline in diastolic function and concurrent changes in myocardial deformation indices, which commonly precede systolic dysfunction. Importantly, these changes may be persistent in postpartum period, particularly in GDM women with obesity and uncontrolled diabetes. The authors could mention and discuss the following papers: PMID: 32530101, PMID: 33949740 and PMID: 37951113.”

Response: Thank you. We have now added to the Discussion: “We have previously reported that at 35–36 weeks' gestation patients with GDM, compared to controls with uncomplicated pregnancies, have reduced left ventricular diastolic and systolic functional indices; additionally, when the cardiac function of these women was assessed at 6 months postpartum, there was a lower degree of improvement in left ventricular myocardial relaxation and in global longitudinal systolic strain despite optimal glucose management during pregnancy [26]. In another study, we found that in women with GDM examined at 26-40 weeks of gestation there is evidence of subclinical reduction in biventricular systolic function [27].

Although in this study we did not include echocardiographic assessment, identifying patients at risk for postpartum dysglycaemia and metabolic syndrome after GDM, would allow to evaluate cardiac function to prevent short- and long-term consequences. Additionally, further larger interventional studies should be carried out to understand the implications of GDM, postpartum dysglycaemia and cardiac remodeling.”

  1. Aguilera J, Sanchez Sierra A, Abdel Azim S, Georgiopoulos G, Nicolaides KH, Charakida M. Maternal cardiac function in gestational diabetes mellitus at 35-36 weeks' gestation and 6 months postpartum. Ultrasound Obstet Gynecol. 2020;56:247-254.

  1. Company Calabuig AM, Nunez E, Sánchez A, Nicolaides KH, Charakida M, De Paco Matallana C. Three-dimensional echocardiography and cardiac strain imaging in women with gestational diabetes mellitus. Ultrasound Obstet Gynecol. 2021;58:278-284.

Kypros H. Nicolaides

Fetal Medicine Research Institute, King's College Hospital

16-20 Windsor Walk, Denmark Hill, London SE5 8BB

email: kypros@fetalmedicine.com

Reviewer 2 Report

Comments and Suggestions for Authors

Author Response

Responses to comments of the reviewers

We enclose a revised manuscript and responses to the constructive comments of the reviewers.

REVIEWER #2

“Overall, this is a sound study evaluating prevalence of cardiometabolic and diabetic status in PP GDM women. The authors should evaluate the data to expand on the findings in the discussion possibly included fetal macrosomia to make more of a connection between the data and the objectives of the study. Is there any correlation to fetal weight at birth with PP diabetic status.

  • Over all the paper is well written, there are a few typographical errors that should be corrected.

Response: Thank you. These have now been corrected.

  • Define dysglycaemia and metabolic syndrome in this section. How common are these within the GDM population and what is the prevalence rate postpartum?

Response: Thank you. The definition of dysglycaemia is provided in Material and Methods (Section – Outcome measures; paragraph 1; lines 85 to 93).  We have identified an error in the definition of dysglycaemia, which has now been corrected: “Prediabetes was defined according to the criteria of the National Institute for Health and Care Excellence (NICE) [13], as HbA1c between 42 and 47 mmol/mol or fasting plasma glucose between 5.5 and 6.9 mmol/L or impaired glucose tolerance (2h blood glucose during OGTT) between 7.8 and 11 mmol/L

Similarly, the definition of metabolic syndrome, is provided in Material and Methods (Section – Outcome measures; paragraph 2; lines 95 to 100).

The definition and prevalence of dysglycaemia postpartum and a new reference have now been added to the Introduction (paragraph 1, lines 38-41).

  1. Cheung NW, Rhou YJJ, Immanuel J, Hague WM, Teede H, Nolan CJ, Peek MJ, Flack JR, McLean M, Wong VW, Hibbert EJ, Kautzky-Willer A, Harreiter J, Backman H, Gianatti E, Sweeting A, Mohan V, Simmons D. Postpartum dysglycaemia after early gestational diabetes: Follow-up of women in the TOBOGM randomised controlled trial. Diabetes Res Clin Pract. 2024;218:111929.

The prevalence of dysglycaemia has also been reported in the Discussion section (lines 216-217): “A meta-analysis of 15 studies of postpartum assessment in 4560 women with previous GDM reported that the incidence of dysglycaemia ranged from 9 to 63%”

  • What are the predictors of the two objectives (dysglycaemia and metabolic)? How are these selected? Discuss.

Response: Thank you. The predictors of both objectives are mentioned in Results (paragraph 3; lines 146-168). In the statistical analysis, we have clarified the explanation of the prediction model and how the variables were selected, as follows (Section Material and Methods, Statistical analysis; lines 112-116).

  • Lines 35-36. The phrasing is unclear. Are the authors suggesting that shortly after delivery glucose levels return to normal in all women or women with GDM? I think the authors in this sentence mean to say that “shortly after delivery glucose levels return to normal in women with GDM, but women with GDM have a 10-fold risk of developing T2DM.” Please clarify this sentence.

Response: Thank you. The mentioned sentence was clarified as follows (Introduction, paragraph 1, lines 36-39): “Shortly after delivery, glucose homeostasis is restored to non-pregnancy levels, but women who had GDM have a ten-fold higher risk for future dysglycaemia (defined as having prediabetes or type 2 diabetes (T2D)) than women with normoglycaemia in pregnancy [3].”

  • Lines 41-42. The authors should provide citation to support why the authors state that GDM women develop T2DM earlier in life.

Response: Thank you. This statement is included in the article referenced as 7 [Tobias DK, Stuart JJ, Li S, Chavarro J, Rimm EB, Rich-Edwards J, et al. Association of History of Gestational Diabetes With Long-term Cardiovascular Disease Risk in a Large Prospective Cohort of US Women. JAMA Intern Med. 2017;177:1735-42].

  • Studies suggest that pregnancy and postnatal weight change influence maternal mortality. The study team could show if pregnancy weight gain and postnatal weight change influenced glycemia and markers of metabolic syndrome.

The weight gain of our cohort when comparing both groups was included in the analysis but showed protective effect for the dysglycaemia group (most likely influenced by clinical intervention). To avoid excessive information in the tables, it wasn’t mentioned.

The median weight gain between 12 and 36 weeks of gestation in the normoglycaemia group was 7.4 (4.1-10.5) kg and in the dysglycaemia group it was 6.3 (3.2-10.0) kg (p=0.006).

  • The authors could report other chronic diseases of enrolled patients and determine if cardiovascular disease, chronic kidney disease, lupus, etc. were associated with postnatal metabolic syndrome and glycemic control.

Response: Thank you. Chronic diseases of the women included in our cohort were part of our database. However, the prevalence of these chronic diseases was very low which did not allow us to fit regression models and draw any conclusions.

  • Table S1. Baseline characteristics. Line 187: under age and prediabetes, correct the age range.

Response: Thank you. This has now been corrected.

  • In the tables, BMI at 11-13 weeks’ gestation and BMI at 36 weeks gestational could be placed together, in ascending order, to make it more visible.

Response: Thank you. These changes have now been made in tables 1, 3, 4, S1 and S2.

  • Getting through all the tables is quite tedious could they be condensed are they all necessary since most of the results are discussed in the results?

Response: Thank you. We have tried simplifying the tables by deleting some rows (mean, IQR) and the tables 3 and 4 have been condensed by deleting references of binary variables.

  • Line 190. Remove the additional period at the end of the last sentence.

Response: Thank you. The additional period has now been removed.

  • Line 203. Are the authors reporting postnatal BMI as the postnatal maternal weight? If so, in the sentence change “weight” to BMI. The authors need to report maternal weight change in the prenatal and postnatal periods and determine whether weight change during these periods influences glycemic control and metabolic syndrome.

The following paragraphs have been modified as there was a typing error:

 Line 165: “When performing multivariable analysis, predictors of metabolic syndrome were need for metformin for glycaemic control of GDM (OR 2.17; 95%CI 1.06-4.42), maternal BMI at the postpartum visit (OR 5.49; 95%CI 2.60-11.59) and absence of breastfeeding (OR 2.14; 95%CI 1.21-3.77).”

Line 212: “Predictors of metabolic syndrome were need for insulin therapy for glycaemic control of GDM, maternal BMI at the postpartum visit and absence of breastfeeding.”

Additionally, this has been corrected in the abstract section (line 27): “Significant predictors of metabolic syndrome were postpartum maternal BMI (OR 5.49; 95%CI 2.60-11.59) and absence of breastfeeding (OR 2.14; 95%CI 1.21-3.77).

As mentioned previously, the weight gain of our cohort when comparing both groups was included in the analysis but showed protective effect for the dysglycaemia group (most likely influenced by clinical intervention). Therefore, and to avoid excessive information in the tables, it wasn’t mentioned.

The median weight gain between 12 and 36 weeks of gestation in the normoglycaemia group was 7.4 (4.1-10.5) kg and in the dysglycaemia group was 6.3 (3.2-10.0) kg (p=0.006).

  • Line 154. Early gestational age at delivery (<24 weeks….) was protective of dysglycaemia. Not sure how relevant this is if the patient didn’t make it to late gestion in this study.

Response: Thank you. This sentence has been typed incorrectly. It should read: “Early gestational age at diagnosis (<24 weeks as defined at the first summit discussing gestational diabetes diagnosed early in pregnancy [15]) was a risk factor for dysglycaemia (OR 3.05; 95%CI 1.90-4.91).

Discussion:

  • Line 278. Change 5-month postpartum to indicate that this study was completed at the median follow-up 5-month postpartum visit. It reads as though this is when they develop GDM.

Response: Thank you. The sentence has now been corrected to: “A high proportion of women who had developed GDM during pregnancy, show evidence of dysglycaemia, at a median of 5 months postpartum follow-up”.

  • The postpartum visit date is not limited to specifically 5 months postpartum. The authors need to re-title this manuscript, expanding the time frame of the study. The postpartum measurements occurred at the median of 5 months, but the IQR was 4.4-7 months. Additionally, in the title “previous gestational diabetes” reads as if all women within this study had GDM in a pregnancy that was previous to the pregnancy recorded in this study. The word previous should be removed from the title or better explained

Response: Thank you. The title has been replaced by “Predictors of cardiometabolic health a few months postpartum in women who had developed gestational diabetes”.

  • In this discussion, these findings below could be expanded upon:

  • Although women who develop GDM tend to have higher maternal age, the mean age of women in this study seems high (>35).

Response: Thank you. Further discussion on maternal age has been added to the Discussion section (paragraph 4, lines 231-237). Therefore, a new reference has been added:

  1. Chen, LW., Soh, S.E., Tint, MT. et al. Combined analysis of gestational diabetes and maternal weight status from pre-pregnancy through post-delivery in future development of type 2 diabetes. Sci Rep 2021; 11: 5021. https://doi.org/10.1038/s41598-021-82789-x

  • “Additionally, in the dysglycaemia 198 group, the prevalence of metabolic syndrome was substantially higher than in the normo-199 glycaemic group (20% vs 7%, p<0.001).”

Response: Thank you. The prevalence of metabolic syndrome has been mentioned in the Discussion section (paragraph 6, lines 245-251). However, further discussion has been added to this paragraph as per reviewer’s suggestion.

Are there any therapies that are being used to address either syndrome (or in general) in GDM postpartum women? Discuss.

Response: Thank you. A paragraph on pharmacotherapy has been added to the discussion (lines 286-292). Therefore, new references have been added:

  1. Ratner RE, Christophi CA, Metzger BE, et al. Prevention of diabetes in women with a history of gestational diabetes: effects of metformin and lifestyle interventions. J Clin Endocrinol Metab 2008; 93(12): 4774-9.

  1. Xiang AH, Peters RK, Kjos SL, et al. Effect of pioglitazone on pancreatic beta-cell function and diabetes risk in Hispanic women with prior gestational diabetes. Diabetes 2006; 55(2): 517-22.

  1. Elkind-Hirsch KE, Seidemann E, Harris R. A randomized trial of dapagliflozin and metformin, alone and combined, in overweight women after gestational diabetes mellitus. Am J Obstet Gynecol MFM 2020; 2(3): 100139.

  1. Elkind-Hirsch KE, Shaler D, Harris R. Postpartum treatment with liraglutide in combination with metformin versus metformin monotherapy to improve metabolic status and reduce body weight in overweight/obese women with recent gestational diabetes: A double-blind, randomized, placebo-controlled study. J Diabetes Complications 2020; 34(4): 107548.

  • Overall, the results should be more of discussion in this section and not just a restatement of the statistical results. It may be beneficial to review demographic data to determine the postpartum diabetic medication being used in cohorts, and to determine if this impacts the two objectives of the study.

Response: Thank you. A paragraph in the Discussion section (paragraph 6, lines 236-244) has been added. Therefore, a new reference has been added.

  1. Golubic R, Car J, Nicolaides K. Enhancing postpartum cardiometabolic health for women with previous gestational diabetes: Next steps and unanswered questions for pharmacological and lifestyle strategies. Diabetes Obes Metab. 2025 Feb;27(2):447-449.

Conclusions:

  • In the conclusions or strengths and weaknesses, the authors need to discuss the impact that elevated maternal age could have on postnatal glycemic control and metabolic syndrome and the correlation with fetal macrosomia.”

Response: Thank you. Age as a risk factor for postnatal dysglycaemia and metabolic syndrome is discussed in the Discussion, subsection on comparison with other studies. We have added a sentence in the Discussion section (paragraph 4, line 237): “Maternal age has also been associated with foetal macrosomia.”

Kypros H. Nicolaides

Fetal Medicine Research Institute, King's College Hospital

16-20 Windsor Walk, Denmark Hill, London SE5 8BB

email: kypros@fetalmedicine.com